# Mixed-Methods Evaluation of the HealthyWEY E-Learning Toolkit for Promoting Healthy Weight in the Early Years

**DOI:** 10.3390/ijerph22020137

**Published:** 2025-01-21

**Authors:** James E. Harrison, Julie Abayomi, Shaima Hassan, Lawrence Foweather, Clare Maxwell, Deborah McCann, Sarah Garbett, Maria Nugent, Daisy Bradbury, Hannah Timpson, Lorna Porcellato, Marian Judd, Anna Chisholm, Nabil Isaac, Beth Wolfenden, Amy Greenhalgh, Paula M. Watson

**Affiliations:** 1Physical Activity Exchange, Research Institute for Sport and Exercise Sciences, Liverpool John Moores University, Liverpool L3 2EX, UK; l.foweather@ljmu.ac.uk (L.F.); inspiredbyfullcircle@icloud.com (D.M.); sgarbett16@outlook.com (S.G.); paula@madeuptomove.co.uk (P.M.W.); 2Faculty of Health Social Care and Medicine, Edge Hill University, Ormskirk L39 4QP, UK; abayomij@edgehill.ac.uk; 3Department of Primary Care and Mental Health, University of Liverpool, Liverpool L69 7ZX, UK; s.m.hassan@liverpool.ac.uk; 4NIHR Applied Research Collaboration ARC NWC, Liverpool L3 5TF, UK; 5School of Public and Allied Health, Liverpool John Moores University, Liverpool L2 2ER, UK; c.maxwell@ljmu.ac.uk (C.M.); l.a.porcellato@ljmu.ac.uk (L.P.); 6Blackburn with Darwen Council, Blackburn with Darwen, Blackburn BB1 7DY, UK; maria.nugent@blackburn.gov.uk (M.N.); beth.wolfenden@tameside.gov.uk (B.W.); amy.greenhalgh@blackburn.gov.uk (A.G.); 7Manchester and Salford Pain Centre, Salford Royal Hospital, Northern Care Alliance, Salford M6 8HD, UK; daisy.bradbury@nca.nhs.uk; 8Public Health Institute, Liverpool John Moores University, Liverpool L2 2QP, UK; h.timpson@ljmu.ac.uk; 9HCRG Care Group Services Limited, Salisbury SP4 6AT, UK; marian.judd@hcrgcaregroup.com; 10Department of Psychology, Institute of Population Health, University of Liverpool, Liverpool L69 3GF, UK; anna.chisholm@liverpool.ac.uk; 11Cornerstone Practice and Health Care, Blackburn with Darwen, Blackburn BB1 2HR, UK; nisaac@nhs.net

**Keywords:** physical activity, obesity, diet, early years, maternity, acceptability

## Abstract

Despite being well-positioned to promote healthy lifestyles in young children, early years practitioners often face barriers to supporting child weight management. This mixed-methods study aimed to assess the preliminary effectiveness and acceptability of an e-learning toolkit (HealthyWEY) designed to upskill and support multi-agency professionals to promote healthy weight in early childhood. A total of 54 health visitors/community nursery nurses, 38 children’s centre staff and 17 other health professionals engaged with the HealthyWEY e-learning, which drew on self-determination theory and consisted of nine modules that were completed over 7–10 weeks. Non-parametric statistical analysis using Wilcoxon’s signed-rank tests were used to explore participants’ practice-based knowledge, psychological needs satisfaction and motivations for prioritising pre-school child weight from pre- to post-intervention. Focus groups (*n* = 11) were conducted with a sub-sample of multi-agency professionals (*n* = 39) to explore the process of implementation across sites, while interviews were also conducted with two parents/carers who took part in consultations with HealthyWEY-trained practitioners. After completing the HealthyWEY e-learning, participants perceived fewer barriers to pre-school child weight management (median change = −0.7; *p* < 0.001), greater autonomy (median change = 0.7, *p* < 0.001), competence (median change = 0.8, *p* < 0.001) and relatedness (median change = 0.4, *p* < 0.001) and a higher autonomous motivation towards promoting healthy weight (median change = 0.3, *p* < 0.001). E-learning was perceived to be highly relevant to participants’ roles and congruent with local child weight strategies. Challenges to implementation included time constraints and disruptions related to the COVID-19 pandemic. Recommendations for a better user experience were offered by enhancing the toolkit’s design and interactivity. Engagement with the HealthyWEY e-learning led to promising changes in perceived barriers and motivational variables. The toolkit was perceived to be acceptable amongst multi-agency workforces, albeit challenging to prioritise within time-pressured health and early years settings.

## 1. Background

With an estimated 40 million children under the age of 5 living with overweight or obesity worldwide, the promotion of healthy weight in early years is crucial [1]. In England, childhood obesity rates are among the highest in Europe, with approximately 1 in 4 children aged 4–5 years classified as overweight (12.2%) or obese (9.2%). Prevalence is highest among those from low socio-economic and ethnic minority backgrounds [2]. Childhood obesity often persists into adulthood and is associated with significant physical and psychological health consequences in later life. Therefore, obesity prevention efforts are warranted to help reduce the risk of chronic health conditions related to both cardiovascular and psychological morbidities, whilst also promoting healthy growth and development during critical early years.

To combat childhood obesity, the concept of early programming has become crucial. It refers to how exposure to external factors during the early development can permanently influence future health [3]. The first 1000 days of life—from conception to a child’s second birthday—are identified as a critical window to promote maternal and child health [4]. During this period, external factors such as high maternal pre-pregnancy weight, high infant birth weight and rapid weight gain, infant feeding practises and the early introduction of solid foods are linked to childhood overweight and obesity [4]. This understanding has led to an increase in interventions targeting these risk factors during pregnancy, thereby implementing early programming strategies to prevent childhood obesity effectively [3]. Maternity and the early years stages thus present an opportune moment to engage families and address modifiable risk factors for early age excess weight gain [5].

Current guidance on managing pre-school weight emphasises the delivery of lifestyle interventions targeting external risk factors including dietary, physical activity (PA) and behavioural components [6]. Consistent with this guidance, previous multicomponent interventions have intervened at the individual, interpersonal and community levels of the socio-ecological model (SEM) [7] with strategies to increase PA, reduce sedentariness and improve the dietary habits of pre-school children and their families [8,9]. Despite previous interventions reporting significant anthropometric changes among pre-school children in the short-term, the absence of continued, long-term strategies is likely to explain the lack of sustained improvement in body composition that has been observed over time; thus, highlighting the need for more sustainable approaches to managing pre-school weight [10].

Despite their short-term effectiveness, implementing multicomponent approaches to promote healthy child weight can be challenging in practice [11]. Early years frontline practitioners often face barriers such as a perceived lack of knowledge and confidence when addressing children’s weight. Organisational obstacles like underdeveloped referral pathways and a lack of capacity due to competing work priorities exacerbate these challenges [12]. Addressing these issues requires targeted workforce training to enhance the skills, knowledge and confidence of practitioners working with young children in health, social and education settings (who, for the purposes of this article, we shall refer to as early years practitioners [EYPs]). Indeed, EYPs have identified specific training needs, including body mass index (BMI) centile calculation and interpretation, weight-related communication and strategies for promoting pre-school nutrition and PA [13].

To optimise the effectiveness of training packages supporting EYPs, it is essential to heed recommendations for developing and evaluating complex interventions that are ecologically valid and meet the needs of end-users [14]. For these solutions to be applicable in ’real-world’ settings, complex intervention research necessitates early engagement with patients, practitioners and policymakers to ensure interventions are acceptable, implementable, scalable and transferable across contexts [15]. Comprehensive guidance provided by the Medical Research Council (MRC) advocates a phased approach to research on complex interventions. This includes the identification and development of the intervention itself, followed by feasibility, evaluation and implementation phases. This systematic process of reflection and action provides a platform for researchers to translate theoretical knowledge and scientific evidence in a way that renders it operational in clinical and public health practice [16].

Between 2015 and 2020, our multi-disciplinary team co-developed and piloted the Healthy Weight in the Early Years (HealthyWEY) e-learning toolkit, which aimed at equipping EYPs with the knowledge and skills for promoting healthy weight in 2–4-year-old children [17]. The toolkit’s online format provided a pragmatic approach to professional development training that offered several advantages over non-internet delivered methods; including flexible, self-paced learning that can be catered for different learning styles and scaled to promote wider engagement among healthcare staff. Co-produced in partnership with staff from local children’s centres, health visiting teams and academics, HealthyWEY integrates self-determination theory (SDT) to foster EYPs autonomous motivation by satisfying needs for autonomy, competence and relatedness [18]. The toolkit synthesised scientific evidence and incorporated current guidelines from organisations including the National Health Service (NHS), the WHO, the NICE and the Royal College of Paediatricians and Child Health (RCPCH), making the information accessible to multi-agency practitioners across early years settings. HealthyWEY was initially piloted in a single geographical location in the UK with single teams of health visitors (including community nursery nurses and public health nurses), general practitioners (GPs) and practice nurses and children’s centre staff. Despite findings suggesting promising changes to EYPs’ perception of the barriers to addressing pre-school weight and increases in their perceived competence and relatedness, the study was small in breadth and limited to the 2–4 years age group [17].

Building on our formative work and recognising the critical period to influence children’s healthy weight management during the first 1000 days of life, HealthyWEY was extended for the current study to encompass pre-conception up to 5 years. The expanded toolkit comprised 9 modules hosted on a micro-site affiliated with the lead university (freely accessible at: https://www.ljmu.ac.uk/Home/Microsites/Promoting-healthy-weight-in-pre-school-children: accessed on 18 November 2024), each module requiring between 15 and 60 min for completion. In addition to its comprehensive content, the toolkit integrated various features and resources to help EYPs consolidate learning, monitor progress and support implementation in their practises. The aim of this study was to evaluate the effectiveness and acceptability of the HealthyWEY e-learning for upskilling a wider and more diverse sample of multi-agency professionals to support healthy weight-related behaviours during infancy and early childhood.

## 2. Methods

### 2.1. Design

Drawing on the MRC’s guidance for developing and evaluating complex interventions, a feasibility study was conducted to assess the effectiveness of the HealthyWEY e-learning toolkit and its acceptability within the context it was implemented [15]. A mixed methods design was used to consider some of the more challenging evaluative questions that emphasised implementation, context and system fit. These questions centred around the fidelity and quality of the implementation (e.g., *what is being implemented and how?*), mechanisms of change (e.g., *how does the intervention produce change?*) and context (e.g., *how does context affect implementation and outcomes?*). This study was approved by the Health Research Authority (HRA), Health and Care Research Wales (HCRW) (*304599*) and the University Ethics Committee (*21/YH/0269*) at the lead institution and was conducted between *November 2021–March 2022*. It is important to highlight that throughout this period, healthcare services across the UK were stretched and capacity was limited as a result of the COVID-19 pandemic.

### 2.2. Participants

The research team identified multi-agency workforces from nine locations across England for geographic and demographic diversity. After virtual meetings with local collaborators, seven of the nine sites agreed to participate. Early years professionals (EYPs) from these sites were invited to virtual introduction events to learn more about the project and consider the value of the e-learning toolkit. Following these events, 206 multi-agency staff consented to join the study. Despite plans to recruit maternity teams from three sites, capacity issues and staff shortages limited participation to health visiting and children’s centre teams.

### 2.3. Intervention

Table 1 provides an overview of the HealthyWEY e-learning intervention using the Template for Intervention Description and Replication (TIDieR) [19].

### 2.4. Outcome Measures

The following measures were implemented to assess the effect of the HealthyWEY e-learning on participants’ knowledge, motivations and perceived barriers to addressing pre-school weight. These measures were assessed via a single questionnaire that participants completed using the online survey platform, JISC on two occasions: at baseline (between *December–January 2021*) and 7–10 weeks post-baseline (between *February–March 2022*; see Appendix A for a copy of the full questionnaire).

#### 2.4.1. Child Weight-Related Practice

Participants were presented with three fictional child weight-related scenarios and asked to provide an assessment of each case (see Figure 1 for an example). Responses to each question were coded against a framework developed by the research team during several meetings that took place during the course of the intervention. Here, marks were awarded for each element of a participant’s response that was deemed to be ‘in line with the HealthyWEY approach’ before being collated to produce a total score. As the questions were open-ended and there were an infinite number of possible marks for each question, it was not possible to define a maximum score. The coding framework was developed by members of the research team (J.H., L.F., J.A., S.H., H.T., L.P. and P.W.) during several meetings in the intervention period. At the end of the intervention, responses to each scenario were coded by a team of three research assistants who were trained by the study’s research officer (J.H.).

#### 2.4.2. Barriers to Addressing Pre-School Child Weight

A 20-item questionnaire adapted from Wu and Steele’s (2011) School Nurses’ Perceived Barriers to Addressing Paediatric Obesity was used to assess participants’ perceptions of the barriers to addressing pre-school weight [21]. Barriers were linked to individual staff (6 items, e.g., not *knowing how to raise a child’s weight-related issue*), individual families (4 items, e.g., families *not perceiving their child’s weight to be a problem*), interpersonal (3 items, e.g., feeling *it is often an inappropriate time to address a child’s weight-related health*) and organisational factors (7 items, e.g., being *unfamiliar with referral routes and pathways*). Responses were scored on a 6-point Likert scale (1 = *not at all a barrier*; 6 = *very much a barrier*) and median values for each of the barriers (e.g., individual staff, interpersonal) were calculated and reported in the subsequent analyses.

#### 2.4.3. Psychological Needs Satisfaction and Motivation

An adapted version of the 18-item Psychological Needs Satisfaction in Exercise Scale was used to assess the perceived autonomy (6 items), competence (6 items) and relatedness (6 items) of participants in addressing pre-school weight [22]. Example questions included ‘*I feel free to make my own decisions when working with families regarding child weight*’ (autonomy), ‘*I feel confident I can do what is required to manage child weight*’ (competence) and ‘*I feel close to my colleagues who appreciate how difficult child weight management can be*’ (relatedness). Responses were scored on a 6-point Likert scale (1 = *false*; 6 = *true*) with higher scores indicating greater needs satisfaction in autonomy, competence and relatedness, respectively. The median values of each sub-scale were subsequently calculated and reported in the analyses.

The 15-item Treatment Self-Regulation Questionnaire was used to assess participants’ motivations for prioritising pre-school weight management [23]. Responses stemmed from the question ‘*the reason I would address child weight issues in my role is...*’ to which responses indicating autonomous motivation (6 items) included ‘*because I personally believe child weight is an important issue*’. Responses indicating controlled motivation (6 items) included ‘*because I feel pressure from others to do so*’ whilst responses pertaining to amotivation (3 items) included ‘*because it’s easier to do what I’m told rather than think about it*’. Responses were scored on a 7-point Likert scale (1 = *not true at all*; 7 = *very true*) with higher scores indicating greater levels of autonomous motivation (i.e., adaptive motivation from within), controlled motivation (i.e., maladaptive motivation driven by internal/external pressures) and amotivation (i.e., a lack of motivation), respectively. As controlled motivation and amotivation scales were measures of negative constructs, a reduced score indicated a positive outcome. The median values of each sub-scale were calculated and reported in the subsequent analyses.

### 2.5. Process Measures

The following measures were used to evaluate the uptake and acceptability of the HealthyWEY e-learning among the participating workforces, as well as the barriers and facilitators to implementation:

#### 2.5.1. Module Completion Log

Whilst completing the e-learning, participants were asked to record their progress by completing a module completion log. Participants were asked to input the dates they started and completed each module, the time taken to complete, whether this was *within* or *outside* of work hours, and any areas where they felt the toolkit could be improved. Participants were asked to return their completion log to the research team electronically on two occasions: half-way through the intervention (i.e., after 4–5 weeks) and at the end of the intervention period (i.e., after 7–10 weeks).

#### 2.5.2. Child Weight Champions Feedback Survey

To supplement the focus groups, all child weight champions were invited to complete a post-intervention feedback survey to explore how the e-learning was implemented within their teams. Here, the champions were asked how they approached the implementation, the strategies they found useful in engaging participants and any challenges they faced. The feedback survey was completed via a single online questionnaire that was accessed through the survey platform, JISC.

#### 2.5.3. Focus Groups

Following the intervention, the research officer (J.H.) conducted semi-structured focus groups with the child weight champions and a subsample of participants from each pilot site. Focus groups with general participants (i.e., those who were not weight champions) were separated by site and by profession (i.e., separate focus groups for health visitors and children’s centre staff from each site; *n* = 7 focus groups in total). Child weight champions were grouped together across localities and invited to attend 1 of 4 dates that the focus groups were scheduled (*n* = 4 in total). The generation of focus group questions were guided by our previous research [17] and the component constructs outlined in the Theoretical Framework of Acceptability for Healthcare Interventions [24], including affective attitude, burden, intervention coherence and self-efficacy, with the aim of exploring the acceptability and feasibility of the e-learning. Questions were based around several key areas with the aim of exploring the acceptability and feasibility of the e-learning. These included participants’ experiences of completing the e-learning, how the training fit within each team’s remit, the effectiveness of the resource for influencing practice, what worked well/less well when approaching the e-learning and how the resource could be improved. In addition, the child weight champions were also asked about any challenges they faced in persuading members of their team to “buy-in” to the e-learning and if so, the strategies they used to overcome this, as well as questions about the implementation process such as challenges, things that worked well and any areas in which they felt this could be improved. In total, 11 focus groups comprising 39 participants (20 weight champions, 17 health visitors and 2 children’s centre staff) were conducted via Microsoft Teams, with each focus group lasting between 40 and 60 min.

#### 2.5.4. Embedded Parent Pilot

As the study’s timeframe was insufficient to have a measurable impact on parental outcomes, an embedded parent pilot was conducted to better understand the impact of HealthyWEY on parental knowledge and confidence to support a healthy weight in their child/ren. At the midpoint of the intervention, participants completing the e-learning over a 10-week period (sites A, C and D) were encouraged to start implementing the skills and knowledge they had gained from the e-learning during consultations with parents/carers. Participants completing the e-learning over a shorter 7-week period (sites B, E, F and G) were given the flexibility to decide whether they participated in the parent pilot, which one additional site (site E) agreed to.

Prior to any consultations taking place, participants from sites A, C, D and E were given a link to an online feedback survey and asked to share this with parents/carers following each consultation. The survey asked parents/carers about their experiences with a HealthyWEY-trained professional, including how they felt during the consultation, how it compared to any previous consultations they may have had and whether they intended to do anything differently at home as a result. At the end of the survey, parents/carers were given the option to attend a 30 min interview with a parent researcher from the HealthyWEY steering group (DM or SG) to discuss their experiences further, which 2 of the 5 parents/carers agreed to.

### 2.6. Analysis

#### 2.6.1. Quantitative Data

Questionnaire data were imported into IBM SPSS 27 for analysis. Data were first assessed for normality using the Kolmogorov–Smirnov test. As the data were found to be not normally distributed, median and interquartile range were reported as measures of central tendency and distribution, respectively, for each outcome measure. Non-parametric statistical analysis was employed in the form of Wilcoxon’s signed-rank tests to explore within-subject differences in participants’ weight-related practises, as well as the perceived barriers, psychological needs satisfaction and motivations for addressing pre-school weight from pre- to post-intervention, with the alpha level for significance set at 0.05.

#### 2.6.2. Qualitative Data

Video recordings from the focus groups and parent interviews were transcribed verbatim into Microsoft Word prior to analysis. Drawing on the guidelines provided by Braun and Clark (2019), transcripts were manually analysed by the research officer (J.H.) using reflexive thematic analysis, which enabled the identification of key themes and sub-themes pertaining to participants’ views of the e-learning; particularly its acceptability, effectiveness and feasibility [25]. This process began with an initial familiarisation of the data through reading each transcript and generating initial codes, which were named as closely to participants’ own words as possible. Thereafter, these initial codes were organised and consolidated into key themes that were perceived to represent participants’ views of the acceptability and feasibility of the e-learning. Participants’ perceptions of the barriers and facilitators to implementation were grouped into the appropriate domains of the Consolidated Framework for Implementation Research (e.g., inner/outer setting, individuals) [20], which seeks to translate and promote implementation theory into meaningful patient care outcomes across multiple contexts.

## 3. Results

### 3.1. Participant Flow Through Study

Figure 2 shows the participant flow through the intervention. Of the 206 multi-agency professionals who consented to take part, 192 participants completed the baseline questionnaire prior to starting the e-learning. During the course of the intervention, 81 participants were lost to follow-up. Of the 111 participants who completed the post-intervention questionnaire, a further 2 were excluded due to missing data. In total, 109 participants were included in the complete case analysis.

At the end of the intervention, 26 of the 33 child weight champions completed the weight champions feedback survey, whilst twenty of the champions also attended the focus groups that took place, with each pilot site having representation from at least one champion. A total of nineteen participants also attended the pilot site focus groups that took place at the end of the interventions, with all but one site having representation from at least one of the participating workforces in their locality.

### 3.2. Participant Characteristics

All 109 participants included in the complete case analysis identified as female and had a mean age of 44.6 years. Of this sample, 35% (*n* = 38) of participants were children’s centre staff, 30% (*n* = 33) were health visitors, 19% (*n* = 21) were community nursery nurses, 13% (*n* = 14) were other health professionals (e.g., public health nurses), 1% (*n* = 1) were maternity staff, 1% (*n* = 1) occupied administrative roles and 1% (*n* = 1) reported their profession as ‘other’. Practice experience ranged from 0 to 41 years, with a mean of 11.6 years. Geographically, 40% (*n* = 44) of participants were based at three sites (B, E and G) in the South of England, 33% (*n* = 36) were based at two sites (A and D) in the North West of England and 27% (*n* = 29) were based at two sites (C and F) in the North East of England.

### 3.3. Outcome Evaluation

#### 3.3.1. Weight-Related Practice

Analysis of the project’s outcome data showed no significant differences in the child weight-related practice of participants from pre- to post-intervention (*p* = 0.610). Participants recorded slightly higher practice-based scenario scores at baseline when compared to post-intervention (see Table 2).

#### 3.3.2. Barriers to Addressing Pre-School Weight

There was a significant decrease in the total barriers to addressing pre-school weight from pre- to post-intervention (*p* = 0.001). Significant reductions were also found in participants’ perceptions of the barriers linked to individual staff, individual family, interpersonal and organisational factors from pre- to post-intervention (*p* < 0.001; see Table 2).

#### 3.3.3. Perceived Psychological Needs Satisfaction Towards Addressing Pre-School Weight

There were significant increases in the perceived autonomy, competence and relatedness of participants from pre- to post-intervention (*p* < 0.001; see Table 2).

#### 3.3.4. Motivations for Addressing Pre-School Child Weight

Participants’ motivations for addressing pre-school weight revealed a small but significant increase in autonomous motivation from pre- to post-intervention (*p* = 0.001) (see Table 2). In contrast, this study found no changes in the controlled motivation or amotivation of participants from pre- to post-intervention.

### 3.4. Process Evaluation

#### 3.4.1. Uptake of the HealthyWEY E-Learning

Analysis of the module completion logs that were returned at the end of the intervention showed that 59% (*n* = 64) of participants completed all nine modules that made up the HealthyWEY toolkit. In contrast, 22% (*n* = 24) of participants completed between one and four modules, 12% (*n* = 13) completed between five and eight modules, 3% (*n* = 3) did not complete any of the e-learning and 4% (*n* = 5) of participants did not return their module completion log at the mid-point or end of the intervention. Of the 101 participants who had completed a minimum of one module, 100% (*n* = 101) and 96% (*n* = 97) completed the two core modules, namely *communicating with parents about child weight* and *behaviour change techniques*, respectively. Additionally, 87% (*n* = 88) and 82% (*n* = 83) of participants completed the two modules relating specifically to pre-school weight, namely *why weight matters* and *assessing weight in young children, respectively,* with 75% (*n* = 76) completing the *infant and child nutrition* module and 72% (*n* = 73) completing the module on *physical activity and sedentary behaviour*. The final three modules, i.e., *nutrition, physical activity and weight during pregnancy*; *roles and responsibilities*; and *cultural considerations*, were the least frequently completed modules with 66% (*n* = 67), 67% (*n* = 68) and 68% (*n* = 69) of participants completing these, respectively. The module completion data appeared to correlate with the order in which the modules were presented in the toolkit, with the modules that appeared earlier in the toolkit having a higher rate of completion than those that featured later in the e-learning.

#### 3.4.2. Child Weight Champions Feedback Survey

A summary of the feedback provided by the child weight champions feedback survey is provided in Table 3. In total, four themes that were consistent features of implementation across each of the seven pilot sites were identified, including the *flexibility to complete the HealthyWEY e-learning*, *monitoring of module completion*, *protected time* and *contact time*.

#### 3.4.3. Focus Groups

Of the 39 participants who attended the focus groups, 64% (*n* = 25) had completed all nine of the HealthyWEY modules, 13% (*n* = 5) had completed between five and eight modules, 18% (*n* = 7) had completed between one and four modules and 5% (*n* = 2) did not return their completion log at the end of the intervention. Four primary themes emerged from the focus groups that related to the acceptability of the e-learning among the health visiting and children’s centre workforces, including *influence on practice*, *suitability*, *changing the weight management culture* and *improvements to the resource*. Each of these are described below, with verbatim quotations attributed to each early years professional to provide illustrative examples of the sub-themes identified.

##### Theme 1—Influence on Practice

Practitioners consistently reported that the e-learning facilitated the development of new skills that could be implemented into daily practice to improve service delivery. Participants referred to the specific modules within the e-learning that they perceived to contain the most valuable information, which included *communicating with parents about child weight* and *behaviour change techniques*:


*“I know I talk about the communication module a lot, but that’s the main one I took so much from. It was just because it was our work really, day-to-day when you’re sat talking, and it’s just changing your way of speaking to them [families with young children] so you’re coming across as you’re there to help them, you’re not there to judge”. *
[Health Visitor, Pilot Site F]

As well as developing skills that were deemed integral to daily practice, participants reported how they felt the e-learning helped to consolidate existing knowledge and provided opportunities for reflection. These themes became apparent with participants commenting on how:


*“it [the e-learning] was informative, it also reminded you of different things that you’d maybe forgotten, but it validated the knowledge that I’d got [Child Weight Champion, Pilot Site C] and “it’s really made me reflect on my practice and how I’m delivering and having those conversations and the language I’m using”.*
[Health Visitor, Pilot Site E]

##### Theme 2—Changing the Pre-School Weight Management Culture

Throughout the focus groups, several participants referred to the sense of personal responsibility they felt the HealthyWEY e-learning promoted when managing pre-school child weight. This was reflected in comments made by a participant who suggested she had:


*“noticed that my position had changed when I was doing the case studies again at the end [of the intervention], because before I was “oh, I’d refer that, I’d refer that” and now I was thinking “yes, I could deal with that and I could answer those queries””.*
[Children’s Weight Champion, Pilot Site C]

The perception that the e-learning promoted personal responsibility among some participants was perceived to coincide with an increase in confidence when managing pre-school weight among. For example, one participant stated:


*“I’m not so intimidated by approaching certain subjects and hopefully, I will have more exposure to working with families that are having these discussions, before I think it is really important”.*
[Children’s Weight Champion, Pilot Site B]

Participants also commented on how they felt the e-learning provided a standardised approach to managing pre-school weight in their locality:


*“it’s the first time we’ve had really detailed e-learning, accessible across the board, so really, really positive and it’s highlighted something that I want to drive forward, that all staff get access to this so we’re all giving the same messages”.*
[Children’s Weight Champion, Pilot Site E]

##### Theme 3—Suitability of the e-Learning

Participants reported the e-learning to be highly relevant to their weight management practises and the remit in which they operated as part of their multi-agency roles, noting the e-learning was helping in addressing the child weight issues they were coming across on a regular basis:


*“we’re seeing a lot more children who are outside their healthy weight pathways as we call it in our service, so this resource [HealthyWEY] and this training is really, really relevant to our practice”.*
[Health Visitor, Pilot Site E]

The suitability of the e-learning for the multi-agency early years workforces was further portrayed as participants reflected on how the training aligned with the current local child weight strategies in their localities. Here, it was claimed that the e-learning:


*“kind of mirrors the current strategy that’s across the borough—[local strategy name] so it fits in really well with that, it’s like a golden thread throughout everything, so that fitted in really well... the HealthyWEY”.*
[Health Visitor, Pilot Site A]

##### Theme 4—Improvements to the Resource

The final theme identified from the focus groups related to the acceptability of the HealthyWEY resource and recommended improvements. Participants commented on the general design of the resource and perceived this as something that could be enhanced moving forward:


*“it [the resource] was a bit clunky, I think, is how I would describe it, in the fact that yes, it didn’t really necessarily flow particularly well when you were trying to move on [through the modules]. It kind of, yes, you had to kind of work your way round it kind of thing”. *
[Health Visitor, Pilot Site E]

By improving the design of the resource, participants felt this would improve the overall user experience, which is an integral component of any e-learning package to ensure those completing the training remain engaged and minimise the perception that this is just another “tick box” activity that individuals are required to complete as part of their mandatory training. The importance of ensuring the toolkit generated a positive user experience was highlighted on several occasions, with participants commenting:


*“I do think all the information that was there does need to be there, it’s just trying to make it [the resource] a little bit more interactive” [Child Weight Champion, Pilot Site A] and “I think it does need to be a little bit more interactive. The videos are really helpful, so yes, like I say, just maybe to implement those”. *
[Health Visitor, Pilot Site F]

###### The Barriers and Facilitators to HealthyWEY Implementation

The themes that were identified from the focus groups relating to the barriers and facilitators to HealthyWEY implementation were grouped into the appropriate domains of the CFIR (e.g., inner/outer setting, individuals) (see Table 4). The factors aiding implementation encompassed the inner level of the framework, whereas the barriers to implementation were identified across multiple settings, including the inner, outer and personal levels. In addition to these themes, the table includes verbatim quotations that provide illustrative examples of participants’ experiences of completing the HealthyWEY e-learning.

#### 3.4.4. Embedded Parent Pilot

A total of five parents/carers consented to take part in the parent pilot that took place at four of the seven pilot sites (A, C, D and E). Geographically, all five participants were from the North-West of England with four identifying as female and one as male. Ethnicity data were provided by all participants, three of whom were white and two were Asian/Asian British. Regarding educational background, three of the five participants had completed a postgraduate degree or other form of postgraduate training, one had completed a foundation degree or equivalent, and one had completed GCSEs/O-Levels.

Data from the parent/carer feedback survey found that each consultation took place at a children’s centre and lasted an average of 30 min. Four consultations were conducted by a children’s centre staff member, with one conducted by a community nursery nurse. Analysis of the feedback survey showed that four of the five participants found their HealthyWEY consultation to be *very helpful*, with one participant finding their consultation *quite helpful*. All participants reported feeling cared for and listened to by the practitioner and agreed that the consultation gave them confidence in managing their child’s diet, weight and/or physical activity. Participants also agreed that their consultation was *much better* than any previous consultation/s they had experienced with other health professionals, with reasons for this centring around the consultations being “*very informative*” [*Parent 1*], “*well explained*” [*Parent 2*] and “*more personal to me and my family which has created achievable goals for us to target*” [*Parent 3*]. When asked if they intended to do anything differently following their consultation, all participants responded with *yes*. These changes included “*increased exercise*” [*Parents 1 and 4*], “*improved eating habits*” [*Parent 2*], and “*better portion control and looking at ways we can add more vegetables into meals and make good sugar swaps*” [*Parent 3*].

After completing the feedback survey, two of the participants signed up to take part in an online interview with a parent researcher (DM/SG) from the HealthyWEY steering group, which gave them the opportunity to provide further feedback following their consultation with a HealthyWEY-trained practitioner and discuss whether this had led to any changes at home. Three themes were identified during the analysis of these interviews, including *parental impact*, *practitioner qualities* and *behaviour change*.

##### Theme 1—Parental Impact

Parents expressed a desire for early access to information regarding children’s diet and physical activity. They felt that having this knowledge from the beginning would have been beneficial, with one parent mentioning:


*“I wish I had had that knowledge [on children’s diet and physical activity] in the beginning when I had my first daughter, and the access to what I’ve got now. I think everyone should have the opportunity to maybe have that information given to you”.*
[Parent 5]

The availability of information and guidance through the consultations also helped parents feel more confident in their parenting choices. As one parent noted:


*“I feel more confident now, because I think you can get lost sometimes, there’s so much information online isn’t there—like when you go on YouTube, on Google”. *
[Parent 5]

Further, parents reported feeling more accountable for their children’s diet and lifestyle choices after the consultations, with one parent explaining,


*“As I say, surprised and informed really, so I went home and sort of now, if anyone tries to suggest giving them something [such as unhealthy food] I’m like ‘well no, he can have this’”.*
[Parent 5]

##### Theme 2—Practitioner Qualities

The way in which practitioners communicated was highlighted as particularly effective during the consultations. Parents appreciated being reassured and understood, with feedback given in a constructive manner. For instance, one parent shared,


*“Also, when she [the practitioner] asked me questions, she used the answer like ‘OK, you are doing this well, but if you do something like this, it can be even better’”. *
[Parent 4]

The practitioners’ approach was described as non-judgemental, which made parents feel more comfortable and open to suggestions. One parent remarked:


*“The way she [the practitioner] came across, she didn’t come across like... ‘judge-y’ she didn’t come across like ‘oh, you’re in the wrong for doing this, you’re in the wrong for doing that’. It’s put across in a way that’s not patronising and not telling you what to do”.*
[Parent 5]

Finally, the health professionals were perceived as being support and helpful; providing advice without making parents feel criticised or personally attacked. A parent commented:


*“I think they’re [the health professionals] more there to help... just to offer advice really, because I think you can take things a bit personally, can’t you?”.*
[Parent 4]

##### Theme 3—Behaviour Change

The consultations led to what both parents described as significant dietary changes at home. They reported reducing the use of processed foods and becoming more mindful of their children’s diet. For example, one parent said,


*“As I say, I’ve stopped using the jars or processed fruits, which I already know the processed fruits, once you blend it in, they release sugars. I have started to do a lot more myself than I would have probably done beforehand”. *
[Parent 5]

Parents also became more conscious of incorporating physical activity into their routines. One parent shared their experience:


*“Now I’ve been more conscious, because obviously I’ve got a double pram, so really, it’s just easier to put her [my daughter] in and it’s quicker to go if I’m in a rush, but now I’ll make sure we can leave a bit earlier to go on a longer walk if we’re going anywhere”. *
[Parent 4]

The overall lifestyle of families seemed to improve as a result of the consultations. Parents recognised the potential for long-term habit formation through small, consistent changes. A parent noted,


*“I think these consultations would help people to potentially...like myself, obviously make lifestyle changes for them and the children, and in turn, once you’ve made these little changes, it’ll help it become a habit”. *
[Parent 4]

## 4. Discussion

This national, multi-centred study evaluated the HealthyWEY e-learning toolkit’s effectiveness in helping multi-agency professionals promote healthy weight in early years. After engaging with the e-learning for 7–10 weeks, participants reported reduced barriers to managing pre-school child weight and increased feelings of autonomy, competence, and relatedness. They also experienced higher motivation and improved practice-based skills, which influenced the weight management culture in their workplaces. Some parents/carers noted enhanced knowledge and confidence after consultations with HealthyWEY-trained practitioners. This study also identified feasibility factors such as autonomy in module selection, drop-in knowledge sharing sessions and support from child weight champions. Barriers included time constraints, competing mandatory training, staff absences, redeployment and COVID-19 disruptions. Feedback highlighted the need for design improvements, increased interactivity, better content flow and progress tracking features.

When addressing the weight status of children, the stigma attached to excess body weight creates multiple obstacles for EYPs to overcome. These barriers include a lack of knowledge on weight management protocols and referral pathways, low confidence in communication skills, uncertainties about identifying excess weight in preschool children and concerns about offending families [5,12]. These challenges highlight the need for increased training to address young children’s weight-related needs [26]. Our observations in the current study draw parallels with previous research that has reported similar changes among EYPs and early childhood educators (ECEs) following an e-learning training intervention [27,28]. Although HealthyWEY is unique in the sense that the e-learning is weight-specific and extends beyond the movement and dietary focus of pre-school children, these comparisons suggest e-learning programmes have the potential to mitigate the perception of barriers that often hinder or prevent discussions around pre-school weight, while strengthening the feelings, attitudes and motivations of EYPs in addressing this complex issue. The preliminary effectiveness of HealthyWEY, along with past research, indicates e-learning’s potential in supporting EYPs, though outcomes may vary due to healthcare context nuances, implementation timelines and evolving practises.

When evaluating the effectiveness of complex interventions in healthcare settings, the significance of the implementation process cannot be overstated [29]. This study illustrates the importance of aligning e-learning interventions with the everyday realities of healthcare environments, where time pressures and competing training priorities can hinder adoption. The successful integration of tools like HealthyWEY may offer a model for other healthcare settings, emphasising the need for adaptive approaches that account for diverse workforce needs, local context and ongoing support systems. The findings suggest that, with the right contextual adjustments, e-learning can be a cost-effective solution to improving early childhood obesity prevention. This process is where theoretical design meets the complex realities of clinical settings, and seamless integration is crucial for success. Contrary to claims that education resources to support EYPs in managing childhood obesity have been inadequate [30], we identified several facilitators for the successful implementation of the HealthyWEY e-learning. These included granting participants autonomy, providing interactive drop-in sessions and identifying local champions to support and oversee implementation [28]. However, challenges such as time constraints, workload and mandatory training took precedence over HealthyWEY and complicated adoption. Additional pressures from staff absences and redeployment during the COVID-19 pandemic, along with the lack of maternity staff participation, further complicated implementation and may explain lower engagement with certain modules (e.g., nutrition, physical activity and weight during pregnancy).

In an endeavour to support EYPs manage the complex issue of preschool child weight and address what has been reported to be a lack of training in behaviour change strategies and effective communication [30], the value of the HealthyWEY e-learning for EYPs was regarded as multifaceted. Participants perceived the modules to offer specialised guidance, equipping them with evidence-based strategies for initiating behaviour change and fostering open, empathetic conversations with parents. The toolkit provided a flexible, self-paced approach, allowing participants to prioritise relevant modules. Additionally, it encouraged collaborative learning among EYPs. Despite these benefits, participants recommended enhancing the toolkit’s design for a more interactive experience by incorporating more quizzes, multimedia elements, real-world case studies and improved content flow. They also suggested progress-tracking features to monitor development and boost motivation to engage with the e-learning.

These findings underscore the critical need for scalable, evidence-based interventions to address childhood obesity at an early stage, as early interventions can prevent the long-term health consequences associated with obesity, such as type 2 diabetes, cardiovascular diseases and psychological challenges in adulthood. Further, this study’s findings also highlight the opportunity for e-learning tools to bridge gaps in training and capacity, particularly for professionals who interact with families in early childhood settings, contributing to a wider public health impact. Policymakers should prioritise integrating weight management and behaviour change strategies into the core training for early years professionals. Ensuring that such training is accessible, engaging and aligned with practical tools like HealthyWEY can reduce professional hesitancy, improve parental interactions and ultimately support healthier weight outcomes for children across diverse socio-economic groups.

### Limitations

While this study has shed light on the HealthyWEY e-learning’s potential to support EYPs in addressing pre-school children’s weight-related needs, there are a number of limitations that warrant consideration. Firstly, the findings are based on the e-learning implementation with health visiting and children’s centre workforces, making it difficult to generalise to other early years workforces due to contextual differences. Notably, the inability to enlist maternity teams meant few professionals worked with pregnant women, which may explain lower engagement with certain modules. The insights gained may not capture the full spectrum of EYPs involved in pre-school weight management. A further limitation that requires consideration are the short-term outcomes that were assessed during the study’s designated timeframe, which naturally imposes a constraint on the extent to which long-term effects and sustained outcomes can be measured. Additionally, 81 participants were lost to follow-up for the outcome data survey, which may have introduced bias and impacted the generalizability of the findings. The loss of these participants may limit the ability to draw definitive conclusions about the overall effectiveness of the intervention, as those who dropped out may differ systematically from those who remained in the study. Future research should aim to improve retention strategies, such as offering additional incentives, increasing follow-up support or employing more frequent touchpoints to reduce participant attrition and ensure a more representative sample for outcome evaluation. Ultimately, pre-school weight management, particularly in a clinical context, is an ongoing process that requires continuous monitoring, intervention and adaptation over an extended period. Due to the study’s short timescale, it was not possible to determine the effectiveness or sustainability of HealthyWEY over time, nor was it possible to capture the evolution of EYPs weight management practises after engaging with the e-learning.

## 5. Conclusions

In summary, this study’s findings provide preliminary evidence of the HealthyWEY toolkit’s effectiveness for supporting EYPs to address the weight-related needs of pre-school children. This study’s findings also supported the acceptability of the e-learning among the multi-agency workforces that took part, with participants recognising the value of the e-learning and the impact that specialised modules around *behaviour change techniques* and *communicating with parents* had on their individual practice. Further research would aid understanding of the long-term impacts of the HealthyWEY e-learning within early years settings, with a focus on sustained outcomes. Future studies could explore how policy changes, such as integrating obesity prevention training into national early years curricula, might enhance the effectiveness of e-learning interventions. Additionally, investigating the scalability of the HealthyWEY toolkit in diverse global contexts and its adaptability to other public health priorities (e.g., mental health, nutrition) could provide valuable insights for large-scale implementation. Further research would aid understanding of the long-term impacts of the HealthyWEY e-learning within early years settings, with a focus on sustained outcomes. Additionally, evaluating the implementation of HealthyWEY with a more diverse range of EYPs (e.g., midwives), primary care (e.g., GPs, nurse practitioners) and secondary care staff (e.g., dietitians, paediatricians) during a time where there are fewer COVID-related barriers (e.g., reduced pressures on services, fewer staff absences) may provide clearer estimates of uptake and effectiveness.

## Figures and Tables

**Figure 1 ijerph-22-00137-f001:**
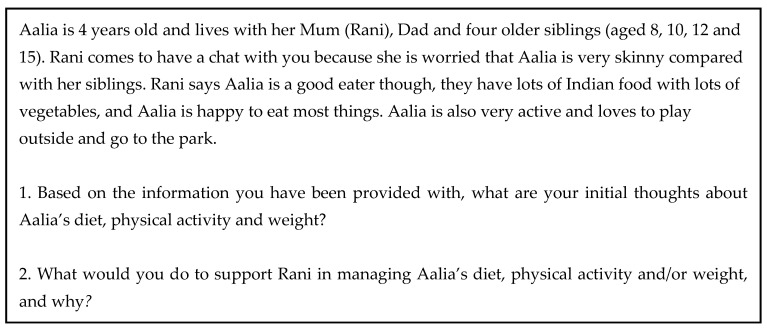
An example of a fictional child weight-related scenario that participants were asked to provide an assessment of pre- and post-intervention.

**Figure 2 ijerph-22-00137-f002:**
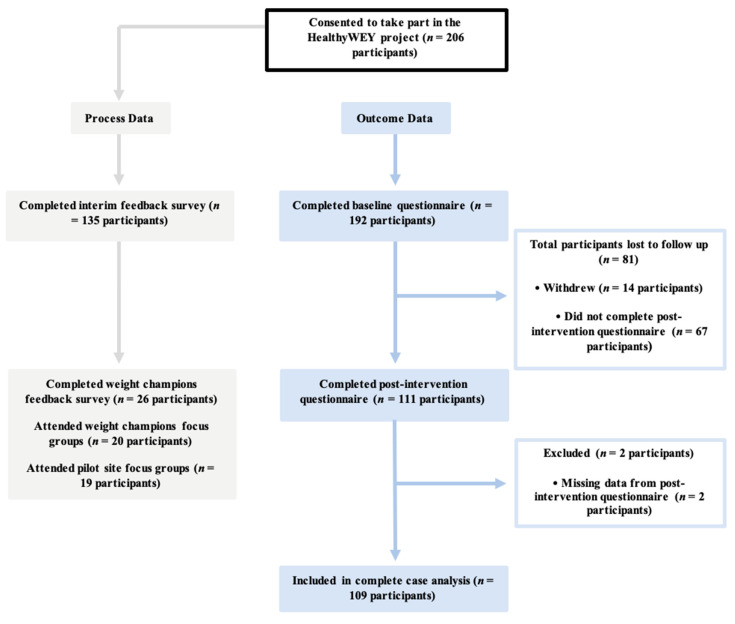
Participant flow through the HealthyWEY intervention.

**Table 1 ijerph-22-00137-t001:** An overview of the HealthyWEY e-learning toolkit using the Template for Intervention Description and Replication (TIDieR; Hoffman et al., 2014). Item 12 of the TIDieR checklist (How Well—Actual) is omitted from the table since this detail is reported in the Section 3 below.

TIDieR Checklist Item	Indicative Content
Item 1. Brief Name	Healthy Weight in the Early Years (HealthyWEY)—an e-learning toolkit for maternity, health visiting and children’s centre workforces to promote a healthy weight in the early years.
Item 2. Why	An e-learning toolkit allowed for a pragmatic and flexible approach to professional development training over conventional training methods. This flexibility allowed individuals to cater the information to their preferred learning styles whilst enabling the e-learning to be scaled to promote wider engagement among healthcare staff during times where capacity and workload may have prevented attendance and/or engagement at non-remote training events. Giving the training needs advanced by healthcare professionals in previous research, HealthyWEY drew on SDT [17] with the aim of facilitating practitioner behaviour change by supporting their need for autonomy, competence and relatedness, whilst also equipping practitioners to support behaviour change in the families of pre-school children.
Item 3. What (Materials)	The HealthyWEY toolkit comprising nine online modules: 1. Communicating with parents about child weight; 2. behaviour change techniques; 3. why weight matters; 4. assessing weight in young children; 5. infant and child nutrition; 6. physical activity and sedentary behaviour; 7. nutrition, physical activity and weight during pregnancy; 8. cultural considerations; and 9. roles and responsibilities. See Appendix A for further information about each module, while the toolkit is freely accessible at: https://www.ljmu.ac.uk/Home/Microsites/Promoting-healthy-weight-in-pre-school-children (accessed on 18 November 2024) In addition to these nine modules, the toolkit also included the following features, all of which can be accessed through the resources tab in the toolkit: Video resources that provide illustrative examples of the strategies/approaches practitioners are encouraged to adopt when discussing children’s weight, nutrition and PA with families. Quizzes and interactive activities to consolidate learning. Bespoke handouts to support implementation in practice (including implementation guidance, information sheets planning tools and module completion log to track progress). A resource bank containing links to other relevant sites and academic references should practitioners wish to consult original evidence sources. Readers should note that the information under the ‘Implementation Guidance’ subheading of the resources tab of the toolkit was not available during the current study. This has been added since and draws on the findings of this research. Participants in the current study did however receive implementation training via a live online workshop that covered similar topics (see Item 4.).
Item 4. What (Procedures)	Local collaborators at each pilot site were asked to recruit a minimum of two ‘child weight champions’ from each participating workforce, who were responsible for overseeing the implementation of the e-learning within their teams and were given the flexibility to decide how the training would be integrated. To facilitate this, the champions were invited to attend a virtual implementation training workshop hosted by the research team. Drawing on the Consolidated Framework for Implementation Research (CFIR) [20] the champions were asked to reflect on the barriers/facilitators to implementation within their inner setting (e.g., team environment), outer setting (e.g., broader political climate) and the individuals taking part (e.g., current knowledge and motivations). The champions then developed a tailored action plan to implement the e-learning in their local area before participants were given access to the HealthyWEY resource to begin the e-learning.
Item 5. Who Provided	The HealthyWEY toolkit was co-produced through multi-stakeholder workshops with health visitors, GPs, children’s centre staff, public health commissioners, parents/carers and academic experts. The toolkit was further adapted for the current study by experts in dietetics (JA), maternity (CM), physical activity (LF) and childhood obesity/behaviour change (PW). Prior to these adaptations being made, the toolkit was reviewed by a group of EYPs and a group of parents/carers of young children, whose feedback fed into these developments. The child weight champions virtual training workshop was led by the project’s principal investigator (PW) and supported by the research officer (JH) and a parent representative (DM, SG) from the research steering group. The implementation of the e-learning was overseen and led by nominated weight champions at each site who had attended the virtual training workshop prior to implementation.
Item 6. How	Participants were able to access the HealthyWEY e-learning through the URL provided under Item 3. Modes of delivery varied between sites, with the children’s weight champions given complete autonomy to implement the e-learning as they deemed fit and the freedom to decide how and when they interacted with participants to discuss progress during this period.
Item 7. Where	Participants were given the option to complete the HealthyWEY e-learning during work hours (e.g., during team meetings permitting time and capacity allowed for this) or at home during their own time.
Item 8. When and How Much	Participants from three of the seven sites (sites A, C and D) completed the e-learning over a 10 week period, whilst those from the remaining four sites (sites B, E, F and G) implemented the training over 7 weeks. Regarding the duration of the HealthyWEY e-learning, each module took anywhere between 15 and 60 min to complete (with some modules being longer than others). All participants were encouraged to complete two “core” modules (communicating with parents about child weight, ~60 min; behaviour change techniques, ~45 min) as a minimum (and before completing the other modules), as these two modules underpin the rest of the toolkit.
Item 9. Tailoring	As described in Item 6, the nominated child weight champions were given the freedom to tailor the e-learning to participants at their respective site as they felt appropriate based on the factors and challenges they identified during the implementation training workshop. For example, at sites where clinical capacity and staff shortages were major challenges, weight champions tailored the intervention by identifying the specific modules they felt were most relevant to their practice and encouraged participants to focus on completing these.
Item 10. Modifications	Due to delays beyond the research team’s control, the intervention period was shortened from the intended 12 weeks and a staggered start was employed. This staggered start was a result of the research team receiving sign-off and approval from each sites’ research departments at different times and explains the discrepancies outlined in Item 8 above.
Item 11. How Well (Planned)	Adherence and fidelity were encouraged with regular emails that the research officer (JH) sent to the children’s weight champions at each site reminding them to encourage staff to continue their progress with the e-learning. Completion of the e-learning was assessed by means of a module log, in which participants were asked to record the date they started and finished a module, the time taken to complete, where they had completed the module and any comments they had (i.e., things they liked, areas for improvement, etc.). Participants were asked to return their module completion log to the project’s research officer (JH) electronically on two occasions: half-way through the intervention (after 4–5 weeks) and at the end of the intervention (after 7–10 weeks). To maintain and/or improve fidelity at each site, the research officer contacted the weight champions half-way through the intervention (once participants had returned their completion logs) to again ask them to remind participants to continue their progress with the e-learning and complete any outstanding modules. At the end of the intervention, fidelity was further assessed during focus groups in which the weight champions were asked how they implemented the HealthyWEY e-learning at their sites.

**Table 2 ijerph-22-00137-t002:** Median (range) pre-test and post-test outcomes data and change scores for participants included in the complete case analysis (*n* = 109).

	Pre-Test Median (Range)	Post-Test Median (Range)	Pre- to Post-Change Median (Range)	Significance *
Practice-Based Scenarios	22.0 (7.0–42.0)	21.0 (7.0–55.0)	−1.0 (−18.0–37.0)	*p* = 0.610
Barriers (Total) ^a^	3.2 (1.0–5.7)	2.5 (1.0–5.6)	−0.7 (−2.9–1.0)	***p* < 0.001**
Barriers (Individual Staff) ^a^	3.0 (1.0–6.0)	2.0 (1.0–6.0)	−0.9 (−4.2–1.2)	***p* < 0.001**
Barriers (Individual Families) ^a^	4.3 (1.0–6.0)	3.3 (1.0–6.0)	−0.8 (−3.8–2.3)	***p* < 0.001**
Barriers (Interpersonal) ^a^	3.0 (1.0–6.0)	2.0 (1.0–5.7)	−1.0 (−3.4–1.7)	***p* < 0.001**
Barriers (Organisational) ^a^	2.7 (1.0–5.1)	2.3 (1.0–5.0)	−0.4 (−2.2–1.3)	***p* < 0.001**
Autonomy Need Satisfaction	3.7 (1.0–6.0)	4.5 (1.0–6.0)	0.7 (−1.3–3.3)	***p* < 0.001**
Competence Need Satisfaction	4.0 (1.3–6.0)	5.0 (1.0–6.0)	0.8 (−2.5–3.2)	***p* < 0.001**
Relatedness Need Satisfaction	5.0 (1.5–6.0)	5.3 (1.0–6.0)	0.4 (−4.0–3.2)	***p* < 0.001**
Autonomous Motivation	6.0 (2.0–7.0)	6.2 (2.5–7.0)	0.3 (−1.8–2.7)	***p* < 0.001**
Controlled Motivation ^a^	3.7 (1.2–5.8)	3.7 (1.0–6.5)	0.1 (−2.2–2.7)	*p* = 0.217
Amotivation ^a^	1.0 (1.0–5.3)	1.0 (1.0–5.0)	0.0 (−2.3–3.7)	*p* = 0.468

* *p* figures in bold reached significance (based on *p* < 0.05). ^a^ Barriers, controlled motivation and amotivation scales measure negative constructs; therefore, a reduced pre- to post-change indicates a positive outcome.

**Table 3 ijerph-22-00137-t003:** Themes derives from the child weight champions feedback survey (*n* = 26) that explored how the toolkit was implemented across the pilot sites and the strategies used to support participants engage with the e-learning.

Themes	Feedback
**Flexibility to Complete E-Learning**	Due to high levels of staff absence and the pressure placed on services as a result of the COVID-19 pandemic at the time, participants at each site were given the flexibility to complete the e-learning at their own place and the autonomy to choose which modules they prioritised.
**Monitoring Module Completion**	To avoid placing additional pressure on staff, module completion was not monitored by the weight champions at any of the seven pilot sites. Instead, participants were encouraged to complete the core modules (1 and 2) as a minimum.
**Protected Time**	To encourage participants to engage with the e-learning, participants at two of the seven sites (A and C) were given protected time to work through the modules. The champions at these sites reiterated how difficult it would have been for participants to complete the e-learning during working hours without this dedicated time. Staff shortages and a lack of clinical capacity prevented participants at the remaining five sites (B, D, E, F and G) being provided with protected time to complete the e-learning.
**Contact Time**	The champions at four of the seven sites (A, C, F and G) made it a priority to meet regularly with colleagues (either virtually or face-to-face) to increase engagement. The champions cited the importance of these meetings for sharing e-learning experiences, reflecting on how the modules could support practice, and offering support where necessary.

**Table 4 ijerph-22-00137-t004:** Themes identified from the focus groups that explored the barriers and facilitators to HealthyWEY implementation with multi-agency early years workforces.

Themes	Subthemes	Illustrative Quotes
Facilitators to implementation —inner setting	Flexibility to complete modules	“*Yes, we were allowed to take...we could either do it within work time if we were able to, and we could take protected time, or we could do it on a day off and be paid for it, so they [the child weight champions] were really supportive in allowing us to take that time to do it where possible”.* [Health Visitor, Pilot Site E]
	Support from child weight champions	“*They [the child weight champions] were really enthusiastic about it, I can recall seeing emails with gentle reminders and just to do what you can type-thing. I knew where the support was, I knew how to access it if I needed it”.* [Health Visitor, Pilot Site C]
	Collaborative learning	“*We were doing modules on the team meeting. I was saying, “oh, I didn’t know about the BMI chart” so it would kind of encourage them [other staff] to then go, “oh right, well that might be something worth learning then” so then they’d go and do it”.* [Health Visitor, Pilot Site A]
	Drop-in sessions	“*I set up twice weekly Teams drop-ins for staff, and just sent out emails saying, “I will be around on...” but I was really pleasantly surprised, actually, that almost all of them [staff] did access those drop-ins”.* [Child Weight Champion, Pilot Site E]
Barriers to implementation —inner setting	Team environment	“*Where a lot of us are working remotely at the moment as well, we’re not all in seeing others face-to-face to be able to have that quick kind of reminder about doing the modules, it’s challenging to just sort of catch up with people”.* [Child Weight Champion, Pilot Site G]
	Other mandatory training	“*We’ve got our mandatory e-learning that we’ve got to do, and that’s all been due recently, so that’s probably played a big part in it, but I think e-learning, people tend to sort of just groan when they hear that word”.* [Health Visitor, Pilot Site B]
	Time and Capacity	“*I think timing and staffing pressures has obviously been the biggest barrier for us, I mean we’ve had staff go to different teams who aren’t even working with us at the moment because they’ve been redeployed elsewhere”.* [Child Weight Champion, Pilot Site F]
Barriers to implementation —outer setting	Staff Absences	“*I think the whole situation with COVID hit a level like I don’t think anybody expected, and it was a challenge just to even do normal service delivery. It eliminated half of our staff”.* [Child Weight Champion, Pilot Site A]
	Business continuity plans	“*Our service ended up going into business continuity over Christmas, and other things had to be prioritised”.* [Child Weight Champion, Pilot Site A]
Barriers to implementation —personal	Staff morale	“*They [the staff] just hadn’t had the chance to even look at it, even look at the emails, let alone anything else, and I think maybe that’s partly because we’re just all frazzled at the moment and we just feel a bit burnt out”.* [Child Weight Champion, Pilot Site B]

## Data Availability

The data underlying this article will be shared upon reasonable request to the corresponding author (j.e.harrison@2022.ljmu.ac.uk).

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
