# Peer review of "Mixed-Methods Evaluation of the HealthyWEY E-Learning Toolkit for Promoting Healthy Weight in the Early Years"

_ijerph, 2025, doi:10.3390/ijerph22020137_

Round 1

Reviewer 1 Report

Comments and Suggestions for Authors

This is a well written article with pertitent information, it was a great read. 

A few suggestions: 

3.2 Participants - If possible, I would include some demographic information on the patients served at the nine sites. Do they serve low-income/non-insuranced, etc..? This would give better insights into adaptability of the program. 

3.8.2 Focus Groups - Please include the methodology behind the generation of the questions. How were the questions generated? Did you look into based literature or was this based on authors experience and what the authors wanted to gather information for? Were the focus group questions reviewed at any point to determine whether edits needed to be made? How did you determine whether data saturation had been reached? 

Results structure - to avoid confusion, I would rearrange the results section to match the methods sections regarding the order to measures are presented. 

Discussion - The study collected a lot of data, but when reading the discussion I felt as though the implications of the results could be fleshed out more. 

Limitations - I would address the 81 participants lost to follow up for the outcome data survey and how that might have impacted your conclusions. 

Author Response

We would like to thank reviewer 1 for their constructive and positive feedback. Our responses to specific comments are provided in the attached document. 

Reviewer 2 Report

Comments and Suggestions for Authors

This is an interested study. This study still need an improvement in some parts. Please find attached our comments in the docs.

Author Response

We would like to thank reviewer 2 for their constructive and positive feedback. Our responses to specific comments are provided in the attached document. 
